# Modeled Early Longitudinal PSA Kinetics Prognostic Value in Rising PSA Prostate Cancer Patients after Local Therapy Treated with ADT +/− Docetaxel

**DOI:** 10.3390/cancers14030815

**Published:** 2022-02-05

**Authors:** Aurore Carrot, Reza-Thierry Elaidi, Olivier Colomban, Denis Maillet, Michel Tod, Benoit You, Stéphane Oudard

**Affiliations:** 1EA3738 CICLY, UCBL-HCL, Faculté de Médecine Lyon-Sud, Université Claude Bernard Lyon 1, 69100 Villeurbanne, France; aurore.carrot@univ-lyon1.fr (A.C.); olivier.colomban@univ-lyon1.fr (O.C.); michel.tod@chu-lyon.fr (M.T.); 2Association Pour la Recherche sur les Thérapeutiques en Cancérologie, 20 Rue Leblanc, CEDEX 15, 75908 Paris, France; relaidi@gmail.com; 3Institut de Cancérologie des Hospices Civils de Lyon (IC-HCL), Oncologie médicale, CITOHL, 69002 Lyon, France; denis.maillet@chu-lyon.fr; 4Centre Hospitalier Lyon-Sud, Department of Medical Oncology, Georges Pompidou Hospital, 20 Rue Leblanc, CEDEX 15, 75908 Paris, France; stephane.oudard@aphp.fr

**Keywords:** prostate-specific antigen, kinetics, prostatic neoplasms, models, theoretical, survival

## Abstract

**Simple Summary:**

In prostate cancer patients with rising prostate-specific antigens (PSAs) after primary local therapy, and at high risk of metastatic disease based on a high Gleason score or ISUP grades 4–5 and/or short PSA doubling time, androgen-deprivation therapy (ADT) was shown to be effective with or without salvage radiotherapy. In the present paper, the analysis of a phase III trial dataset comparing ADT +/− docetaxel demonstrates that longitudinal PSA kinetics during the first 100 days of treatment, assessed using mathematical modeling, is associated with patient prognosis. Indeed, multivariate analyses showed that the modeled PSA production rate constant KPROD, and the elimination rate constant KELIM, exhibited strong and independent prognostic values regarding PSA progression-free survival (PSA-PFS) and overall survival (OS), respectively. As it was shown with the longitudinal CA-125 kinetic in ovarian cancers, the modeled longitudinal PSA kinetics during the 100 days of treatment may represent a novel prognostic factor in patients with metastatic prostate cancers.

**Abstract:**

Background: In metastatic prostate cancer (PCa) patients, androgen-deprivation therapy (ADT) combined with chemotherapy or next-generation androgen receptor targeted agents is a new standard treatment. The objective of the present study is to assess longitudinal PSA kinetics during treatment using mathematical modeling, to identify the modeled PSA kinetic parameters able to exhibit early prognostic/predictive values. Methods: Phase III clinical trial dataset (NCT00764166) comparing ADT +/− docetaxel in 250 locally treated patients for PCa with rising PSA levels, who were at high risk of metastatic disease was assessed. A kinetic-pharmacodynamic (K-PD) model was used to fit PSA kinetics during the first 100 treatment days, to estimate the modeled PSA production rate K (KPROD) and elimination constant rate K (KELIM). The prognostic value of these parameters, considered as categorized (favorable vs. unfavorable) covariates regarding PSA progression-free survival (PSA-PFS) and overall survival (OS), was assessed using univariate/multivariate analyses. Results: Data from 177/250 patients was assessed. KELIM exhibited a significant prognostic value regarding PSA-PFS and KPROD regarding OS (univariate analysis). In the PSA-PFS final multivariate model, KELIM and the primary therapy type were significant. The OS multivariate model integrated both KPROD and baseline PSA doubling-time. Conclusion: In this first study assessing the modeled PSA kinetics prognostic value in PCa patients treated with systemic treatments, KELIM and KPROD exhibited respective prognostic values regarding PSA-PFS and OS.

## 1. Introduction

In patients treated with primary therapy (surgery or radiotherapy) for localized prostate cancer (PCa), biochemical relapses (BR) with the rise in the isolated prostate-specific antigen (PSA) are observed in up to 30% of them within 7 years following the primary therapy [1]. For men who underwent radical prostatectomy (RP) as the primary treatment and subsequently developed PSA recurrence, the main unfavorable prognostic factors were a PSA doubling time (PSA-DT) of 1 year and a pathological Gleason score (pGS) of 8–10 (International Society of Urological Pathology (ISUP) grades 4–5). For patients with a PSA recurrence following primary RP, an interval from primary therapy to biochemical failure (IBF) of <18 months and a biopsy Gleason score (bGS) of 8–10 (ISUP grades 4–5) were the main unfavorable prognostic factors. The EAU prostate cancer guidelines panel recommended the use of a novel BR classification system that stratified patients with BR into low- vs. high-risk categories. After BR in case of RP, salvage radiation therapy can be an option in high-risk patients [2].

For decades, androgen-deprivation therapy (ADT) was considered as a standard treatment in the case of metastatic hormone-sensitive prostate cancer (mHSPC). Over the last 5 years, the treatment landscape has dramatically changed with the addition of 4 systemic agents that previously demonstrated benefits in the castrate-resistant setting [3,4,5,6,7]. In patients with only a biochemical relapse, no systemic treatment was approved [8]. However, the high risk of developing metastases may justify early ADT administration [9], which can be balanced with the long term tolerability of this treatment [10].

In metastatic hormone-naive prostate cancer patients, docetaxel and abiraterone became the treatment options, based on CHAARTED (Chemo Hormonal Therapy versus Androgen Ablation Randomized Trial in Extensive Disease), LATITUDE and STAMPEDE (Systemic Therapy in Advanced and Metastatic Prostate Cancer evaluation of Drug Efficacy) [11,12,13,14]. In addition, recently enzalutamide and apalutamide were added to the armamentarium, after ARCHES, ENZAMET and TITAN phase III studies [15,16,17]. Radiation treatment of the primary in patients with oligometastatic disease recently became an additional option for patients, due to STAMPEDE-arm H. In line with the data suggesting the overall survival (OS) benefit with the addition of docetaxel to ADT in patients with metastatic hormone-naive prostate cancer (GETUG-15, STAMPEDE and CHAARTED trials) [11,12,18], Oudard et al. investigated the utility of this strategy in patients with rising PSA, who are hormone-naive at a high-risk of metastatic progression in a randomized trial. Unfortunately, no benefit in PSA-progression free survival (PSA-PFS) and OS was found with the addition of docetaxel to ADT [19].

Despite this outcome, it is important to search for subgroups of patients who may have benefited from this approach. Consistent with the strategy developed with the modeled kinetics of the serum tumor marker CA-125 in ovarian cancer patients treated with chemotherapy, producing a strong and reliable prognostic marker of OS called KELIM (CA-125 ELIMination constant rate K) [20,21,22], we proposed to assess the prognostic value of PSA kinetics in this trial as a way of identifying some patients who might have benefited from adding docetaxel to ADT.

Of note, the PSA working group has adopted a PSA response criterion, defined as a >50% decrease in PSA with an absolute PSA level lower than 0.2 ng/mL [23]. However, as already found with CA-125 in ovarian cancers, this two time point-based kinetic strategy is largely impacted by the high inter- and intra-individual variability of sparse PSA concentrations measured with different assays, far from the actual dynamics of serum marker kinetics, thereby limiting the comparability in different patient populations than those assessed initially [24].

In that context, mathematical modeling with population kinetic approach of serum tumor marker dynamics during treatment offers several advantages, as it enables to calculate the mathematical equations describing the longitudinal serum tumor kinetics of the population of patients, and then those of individual subjects based on a minimum of three values, with reduced impact of measurement variability. Moreover, it enables to extract a modeled kinetic parameter of interest prone to exhibit prognostic or predictive value, as seen with CA-125 KELIM [20,21,22].

The purpose of the present study is to assess the longitudinal PSA kinetics using mathematical modeling in patients enrolled in the Oudard et al. trial, in order to identify the modeled kinetic parameter of interest (such as KELIM) prone to exhibit prognostic value or predictive value regarding the benefit from adding docetaxel to ADT, in terms of PSA-PFS and OS.

## 2. Materials and Methods

### 2.1. Patients

We analyzed the dataset of the randomized phase III trial (NCT00764166) [19], comparing two arms of treatment (endocrine therapy with ADT, or ADT + docetaxel 70 mg/m^2^ every 3 weeks) in 250 patients with localized PCa previously treated with primary radical prostatectomy (RP) or radiotherapy (RT) and experiencing isolated PSA rising (at least 3 consecutives PSA measurements, such as defined by the American Society for Therapeutic Radiology and Oncology PSA recurrence criteria [25]) with at least one or more high-risk of metastatic disease characteristics (Gleason score ≥ 8, PSA velocity > 0.75 ng/mL, PSA doubling-time (DT) < 6 months, node-positive adenocarcinoma, positive surgical margins, and time to PSA recurrence 12 months or less) [26]. The primary endpoint of the trial was PSA-PFS, while secondary endpoints were PSA response, radiologic PFS, and OS.

To be included in this analysis, a minimum of three PSA assays in the first 100 days of treatment was required to be able to fit individual PSA kinetics to a non-linear model. The patients who had PSA titers rising >10% during the first 100 days of treatment were excluded, as this treatment could not be considered effective, which was inadequate with respect to the objective of the present study.

In addition to patients PSA concentrations and sampling dates, the following clinical data were collected: age at diagnostic, treatment arm, baseline PSA doubling-time (PSA-DT), baseline PSA velocity, the type of primary therapy (RT/RP), pTNM stage, and Gleason score [19].

### 2.2. Mathematical Modeling of PSA Kinetics

Individual PSA data were log transformed to eliminate right-skewness in the data distribution. The mathematical modeling of PSA kinetics in the blood with a non-linear mixed-effect model was the same used from CA-125 in ovarian cancer [21,22], and assessed with the observed PSA values measured during the first 100 days of treatment starting on cycle 1 day 1 of chemotherapy, regardless of treatment arm.

PSA kinetics were described by the PSA production rate KPROD (ng·mL^−1^·day^−1)^, a PSA elimination constant rate KELIM (day^−1^), both being impacted by the indirect effects of treatment with ADT +/− docetaxel (Figure 1).

The semi-mechanistic model structure relied on a central compartment where the blood serum concentration of PSA was described through a production rate KPROD, balanced with an elimination rate constant KELIM. KPROD was regulated by the effects of the systemic treatment, described with a 2-virtual-compartment model (a central compartment C1 receiving chemotherapy dosing and a transit compartment C2) on the cancer cells through an indirect effect, characterized by the half-maximal effective concentration EC50. Because the concentrations of the chemotherapy were not available, a kinetic-pharmacodynamic (K-PD) model was developed to assess the longitudinal kinetics of PSA with the administration dose of chemotherapy arbitrarily set to 1, as already performed for the pharmacokinetics model [27].

### 2.3. Model Qualification

The PSA predictions, obtained with the fitting of data on the K-PD model, were qualified using common algorithms, such as relative standard error (RSE), goodness-of-fit plots (GOF), meaning plots comparing observations vs. predictions, the distribution of normalized prediction distribution errors (NPDE) or visual predictive check (VPC). VPC represented 500 replicates of all individual PSA decline profiles simulated from the final parameter estimates of the model: the 10th, 50th, and 90th percentiles of the observed PSA values were visually included in the 95% Confidence Interval (CI) of simulations. VPC were used as models for internal validation.

### 2.4. Prognostic Value of Modeled PSA Kinetics during the First 100 Days

The prognostic value of modeled PSA kinetic parameters regarding PSA-PFS and OS was assessed using univariate and multivariate survival tests. The modeled PSA kinetic parameters of interest (KPROD and KELIM) were standardized and dichotomized by their median, respectively.

Kaplan–Meier and log-rank tests were used for univariate analyses. All variables found significant with *p* < 0.2 in univariate analyses were integrated in a multivariate Cox proportional hazards regression model. The final Cox regression survival model was obtained using a backward selection. The following potential prognostic factors were also tested in the multivariate analysis: age at diagnostic, treatment arm, baseline PSA doubling time (PSA-DT), baseline PSA velocity, type of previous primary therapy (RT/RP), pTNM stage, and ISUP classification.

Moreover, concordance index (C-index) as a function of the time was used to assess the prediction benefit related to their addition in the multivariate model.

All survival analyses were performed with a 100-day landmark time point analysis, meaning that all patients experiencing events (lost to follow-up, progression or death for PSA-PFS; and lost to follow-up or death for OS) during the first 100 days were excluded. Indeed, the modeled kinetic parameters were estimated during the first 100 days of treatment. This strategy avoided any bias between survival predictions and PSA kinetics parameters estimations [21,22].

### 2.5. Statistical Analyses and Computing Process

All tests were implemented using a two-sided 0.05 alpha risk. The NONMEM 7.5.0 (ICON Development Solutions, Ellicott City, MD, USA) software was used to fit the semi-mechanistic model to PSA kinetics data [28]. Parameters were estimated adjusting data to the model with the First Order Conditional Estimates with Interaction (FOCEI) algorithm plus the Laplacian option. Error model included the PSA Lower Limit of Quantification (LLOQ = 0.1 ng/mL) thanks to the M3 method [29]. Standard Errors (SE) were estimated by 2000 samples bootstrap. Survival analyses and graphical representations of results were performed in R language (R© 3.6.1 software). The Xpose 4.7.1 package in R was used for representation of model qualification (visual predictive check (VPC) and goodness-of-fit (GOF) plots) [30].

## 3. Results

### 3.1. Patient Selection

Of the 250 patients enrolled in the trial, 177 patients were assessable for PSA kinetic modeling, including 94 and 83 patients in ADT + docetaxel and ADT alone arms, respectively. The characteristics of the assessed patients are presented in Appendix A. In summary, a bit more than 50% of patients had previous T3/T4 stage diseases. Most of them were previously treated with prior prostatectomy (68%), and/or radiotherapy (31%). Moreover, the majority of patients had high-risk features at inclusion, based on PSA doubling time and PSA velocity (Appendix A).

The characteristics of patients were well balanced in between treatments arms. The features of excluded patients were similar to those assessed in the present study for most of them, except for baseline PSA and percentage of patients previously treated with radiotherapy, which were slightly lower in the excluded patients.

The median follow-up was 9.8 years, and the median number of PSA values by patients was 4. Of the 177 patients eligible for the PSA kinetics modeling, 173 were assessable for PFS survival analyses, according to the 100-day landmark, including 92 and 81 patients in ADT + docetaxel and ADT alone arms, respectively. No patient presented an OS event before 100 days (Figure 2).

### 3.2. Model Qualification

Parameter estimates from the semi-mechanistic model are presented in the Appendix A. Relative standard errors of parameters and their inter-individual variation, which represent the estimation precision, were lower than 25%, as was the shrinkage (<30%, excepted for K and EC50), thereby suggesting limited risks of biased individual estimates of parameters.

The GOF plots showed that individual PSA profiles were properly fitting observations during the first 100 treatment days by the model. Weighted residuals approximately follow a normal distribution. Moreover, the VPC revealed the correct predictive performance of the model with no aberrant predictions (Appendix A).

### 3.3. KELIM and KPROD Prognostic Values on PSA-PFS and OS in Univariate Analyses

Among the five parameters estimated, KELIM and KPROD prognostic values were considered of interest, as these parameters, meant to characterize PSA production and elimination, were well estimated compared to other modeled kinetic parameters, with lower RSE and lower shrinkage (Appendix A). KPROD and KELIM were standardized by their median, respectively (median KPROD = 0.002 ng·mL^−1^·day^−1^, median KELIM = 0.081 day^−1^), and only the standardized parameters were used in the following analyses.

KELIM and KPROD estimations were not significantly different across treatment arms (KELIM, *p* = 0.39; KPROD, *p* = 0.093) (Figure 3A,B).

KELIM exhibited a significant prognostic value regarding patient PSA-PFS. The median PSA-PFS was 10.1 months in patients with unfavorable KELIM vs. 15.1 months in patients with favorable KELIM (*p* < 0.001) (Figure 4A,B). KELIM was also significant for OS, with a median of 157 months in patients with unfavorable KELIM, compared to a median survival “not reached” in patients with favorable KELIM (*p* = 0.035) (Table 1).

KPROD also exhibited significant prognostic value regarding patient PSA-PFS and OS (Figure 4C,D). The median PSA-PFS was 9.8 months for patients with unfavorable KPROD vs. 15.1 months for patients with favorable KPROD (*p* < 0.001). The median OS was 157 months in unfavorable KPROD patients, and not reached in those with a favorable KPROD (*p* = 0.037) (Table 1).

The other significant prognostic covariates using univariate analyses with a *p* < 0.2 were age at diagnosis (*p* = 0.12), cancer stage (*p* = 0.13), baseline PSA velocity (*p* = 0.18) for PSA-PFS, and age at diagnosis (*p* = 0.052), primary therapy type (*p* = 0.1) and positive node (*p* = 0.198) for OS.

### 3.4. KELIM and KPROD Prognostic Value on PSA-PFS and OS in Multivariate Analyses

#### 3.4.1. Multivariate Survival Models Integrating KELIM as a Modeled PSA Kinetic Parameter

In the final multivariate Cox model regarding PSA-PFS, the following covariates were significant: KELIM (favorable vs. unfavorable, HR = 0.63, 95% CI (0.43; 0.92), *p* = 0.015); and primary therapy type (RP vs. RT, HR = 0.41, 95% CI (0.28; 0.61), *p* < 0.001).

In the final multivariate Cox model regarding OS, the following covariates were significant: KELIM (favorable vs. unfavorable, HR = 0.57, 95% CI (0.43; 0.92), *p* = 0.029); and baseline PSA-DT (<6 vs. ≥6 months, HR = 0.55 (0.33; 0.91), *p* = 0.02) (Table 1).

#### 3.4.2. Multivariate Survival Models Integrating KPROD as a Modeled PSA Kinetic Parameter

In the final multivariate Cox model regarding PSA-PFS, the following covariates were significant: KPROD (favorable vs. unfavorable, HR = 0.65, 95% CI (0.45; 0.96), *p* = 0.028); and primary therapy type (RP vs. RT, HR = 0.42 (0.29; 0.62), *p* < 0.001).

In the final multivariate Cox model regarding OS, the following covariates were significant: KPROD (favorable vs. unfavorable, HR = 0.53, 95% CI (0.3; 0.89), *p* = 0.015); and baseline PSA-DT (<6 vs. ≥6 months, HR = 0.55 (0.33; 0.91), *p* = 0.021) (Table 1).

#### 3.4.3. C-Index Analyses

Looking at the C-index, the combination of KELIM and primary therapy seemed to be the most informative model, rather than KELIM alone or primary therapy alone for the PSA-PFS (Figure 4E).

Regarding OS, the combination of KPROD with baseline PSA-DT was associated with the highest predictive index (Figure 4F).

## 4. Discussion

As for ovarian cancer patients, K-PD modeling was able to characterize the longitudinal PSA dynamics during treatment based on PSA production KPROD, PSA elimination KELIM, and indirect treatment effect on patients, independent of the selected time points used to calculate it. A previous study already suggested that mathematical modeling was a promising tool to assess the longitudinal kinetics of PSA after radical prostatectomy and to predict the risk of early relapse [31].

A major outcome of the present study is the demonstration that modeled PSA kinetics during the first 100 days of systemic treatment inform on the patient prognosis in terms of PSA-PFS and OS. Indeed, patients with favorable KELIM and KPROD experienced significant longer median PSA-PFS and OS in both arms. PSA KELIM was found to be associated with PSA-PFS, but exhibited lower prognostic value regarding OS (contrary to what was reported with CA-125 KELIM in ovarian cancer patients) [21,22]. On the other hand, KPROD seemed to be a stronger prognostic factor regarding OS. Multivariate analyses confirmed that KELIM and KPROD prognostic values regarding PFS and OS were independent, regardless of other prognostic covariates.

Therefore, the present study suggests that KELIM and KPROD might help to identify patients likely to experience shorter or longer survivals.

As KELIM and KPROD values were similar in both treatment arms, these parameters could be calculated in all patients regardless of their current treatment. On the other hand, it could not be used to identify the patients who would benefit from adding docetaxel to ADT. Therefore, it was not a predictive marker of docetaxel efficacy.

Similar outcomes were obtained with the CA-125 modeled kinetic elimination rate constant K (KELIM) in ovarian cancer patients treated with chemotherapy. Indeed, a meta-analysis database of 8 randomized trials involving more than 5000 patients was investigated. KELIM difference between treatment arms was not a surrogate marker of survival benefit in a meta-analysis with more than 5000 patients (Corbaux et al. Proc ESMO 2021). However, consistent with other 7 studies in more than 7000 patients, KELIM was found to be a strong and reproducible prognostic marker regarding PFS and OS, which could be used to discriminate 3 prognostic groups. The poor prognosis group was identified as the subgroup that should be prioritized for innovative treatment meant to reverse the chemoresistance.

As a consequence, additional retrospective analyses on other independent trial datasets will be required to fully understand the prognostic/predictive role of modeled longitudinal PSA kinetic parameters, such as KELIM or KPROD, in patients treated with chemotherapy and/or endocrine therapy, along with prospective validation studies. If the prognostic/predictive value of these parameters was confirmed, they could be easily calculated online at patient bedside on https://www.biomarker-kinetics.org/presentation (accessed on 5 December 2021), where the models can be implemented for routine application.

The limitation of this study is the low number of patients in the trial. This number was reduced further after the selection of patients who experienced a PSA decline in order to obtain a homogeneous population. This strategy was chosen for preventing wrong parameter estimations. However, it might have introduced disbalances in some characteristics of patients and introduced biases in the outcomes.

Cox regression power for OS analysis was 0.38. Our analyses estimated that the number of patients should be doubled in both KELIM/KPROD groups to ensure a standard power of 80%. Thus, our outcomes are mainly hypothesis generating, and the validation of these present results on other PSA databases with larger numbers of patients could reinforce these results.

## 5. Conclusions

This is the first study assessing modeled PSA kinetics in recurrent PCa patients with a rising PSA hormone-naive and at high risk of developing metastasis treated with ADT +/− docetaxel.

Despite limitations due to the low number of patients, the outcomes of the present exploratory retrospective study suggest that modeled kinetic parameters KPROD and KELIM might help to identify recurrent PCa patients with a higher probability of biological progression and survival when treated with ADT +/− docetaxel.

## Figures and Tables

**Figure 1 cancers-14-00815-f001:**
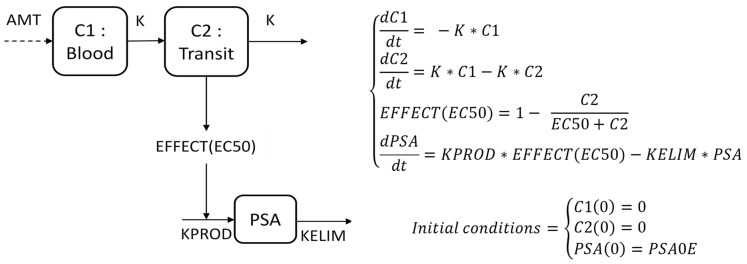
Description of the semi-mechanistic PSA kinetic model. Differential equations and compartment representation of the K-PD PSA kinetic model. AMT (Amount): dose administered, fixed to 1, due to missing pharmacokinetic data. C1 and C2: Concentrations of administered treatment and transit compartment. PSA: PSA compartment. EC50: Treatment concentration required to obtain 50% of the maximal effect (arbitrary unit). K: treatment kinetic rate constant (day^−1^). KPROD: PSA production rate (ng·mL^−1^·day^−1^). KELIM: PSA elimination rate constant (day^−1^). EFFECT: Indirect pharmacodynamic response on production.

**Figure 2 cancers-14-00815-f002:**
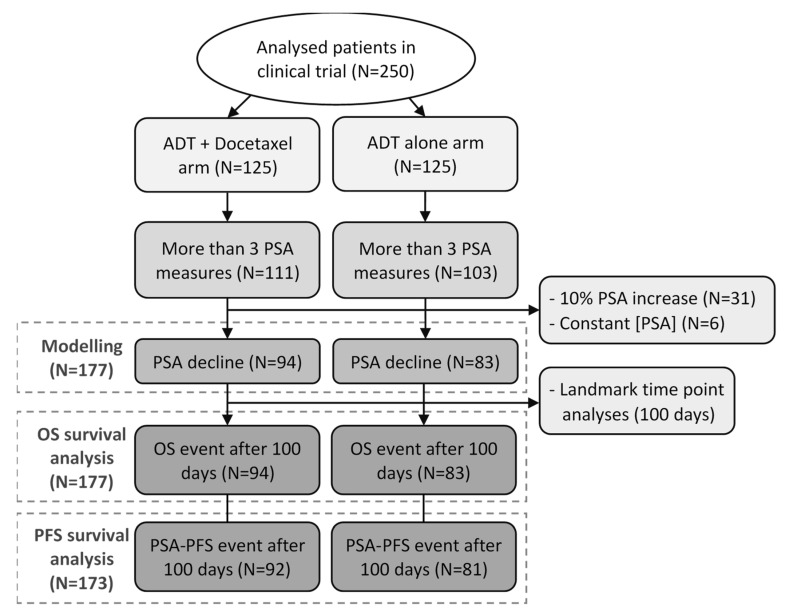
Flowchart of patients selected for the modeling and survival analyses. Patients’ selection included in the present study.

**Figure 3 cancers-14-00815-f003:**
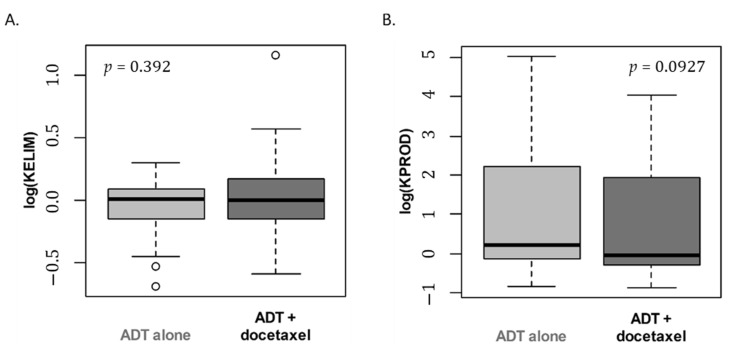
Boxplot of standardized KELIM (**A**) and KPROD (**B**) individual estimations according to treatment arm (logarithm scale). Wilcoxon test *p*-values.

**Figure 4 cancers-14-00815-f004:**
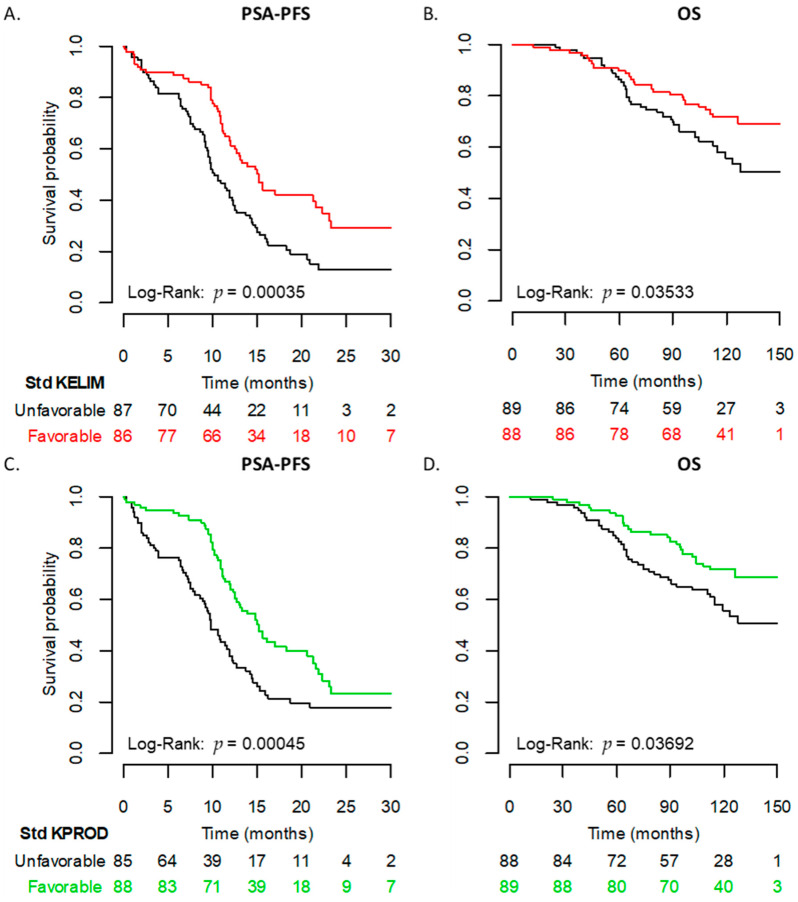
Univariate analyses (Kaplan–Meier) and multivariate concordance index (C-index) analyses. PSA-PFS (**A**) and OS (**B**) Kaplan–Meier curve according to categorical KELIM. PSA-PFS (**C**) and OS (**D**) Kaplan–Meier curve according to categorical KPROD. C-index as a function of the time for PSA-PFS (**E**) and OS (**F**) comparing different Cox regression models.

**Table 1 cancers-14-00815-t001:** Kaplan–Meier and Cox regression results. Kaplan–Meier results (median time in months) for categorical KELIM and KPROD. NR: Not reached. CI: Confidence interval. Cox regression results for categorical KELIM and KPROD. HR: Hazard Ratio. CI: Confidence Interval. PSA-DT: PSA doubling time.

Univariate Analyses (Kaplan–Meier and Log-Rank Test)					
		Median [95% CI]		*p*-Value	
PSA-PFS					
KELIM					
*Unfavorable*		10.1 [9.3; 12.4]		<0.001	
*Favorable*		15.1 [12.6; 22.2]			
KPROD					
*Unfavorable*		9.8 [8.8; 12.1]		<0.001	
*Favorable*		15.1 [12.6; 21.2]			
OS					
KELIM					
*Unfavorable*		157 [115; NR]		0.035	
*Favorable*		NR [NR; NR]			
KPROD					
*Unfavorable*		157 [NR; NR]		0.037	
*Favorable*		NR [115; NR]			
Multivariate Analyses (Cox Regression)					
	HR [95% CI]		*p*-Value		C-Index [95% CI]
Analyses with KELIM					
PSA-PFS					
KELIM					0.66 [0.62; 0.70]
*Unfavorable*	Reference		0.015		
*Favorable*	0.63 [0.43; 0.92]				
Primary therapy					
*Radiotherapy*	Reference		<0.001		
*Prostatectomy*	0.42 [0.29; 0.62]				
OS					
KELIM					0.6 [0.53; 0.67]
*Unfavorable*	Reference		0.02		
*Favorable*	0.55 [0.33; 0.91]				
PSA-DT					
*≥6 months*	Reference		0.029		
*<6 months*	0.57 [0.34; 0.95]				
Analyses with KPROD					
PSA-PFS					
KELIM					0.66 [0.61; 0.71]
*Unfavorable*	Reference		0.028		
*Favorable*	0.65 [0.45; 0.96]				
Primary therapy					
*Radiotherapy*	Reference		<0.001		
*Prostatectomy*	0.42 [0.29; 0.62]				
OS					
KELIM					0.61 [0.54; 0.68]
*Unfavorable*	Reference		0.015		
*Favorable*	0.53 [0.32; 0.89]				
PSA-DT					
*≥6 months*	Reference		0.021		
*<6 months*	0.55 [0.33; 0.91]				

## Data Availability

The data presented in this study are available in this article and Appendix A.

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
