# Peer review of "Modeled Early Longitudinal PSA Kinetics Prognostic Value in Rising PSA Prostate Cancer Patients after Local Therapy Treated with ADT +/− Docetaxel"

_cancers, 2022, doi:10.3390/cancers14030815_

Round 1

Reviewer 1 Report

The authors provide a review of phase III trial and assess the role of PSA kinetics as a prognostic marker for oncologic outcomes. I have a few concerns with this paper:

Major issues

Methods: The authors do not clearly state the method of calculating KPROD and KELIM. Certainly not a sufficient level that would allow replication of the work. Suggest disclosing the equations

Figure 1 is not clear and does not improve my ability to understand the methodology

Overall patient demographics are not described

While the model is complex./comprehensive, I am unsure how this adds value to more conventional measures of PSA kinetics (eg. doubling time). This appears equally as predictive as KPROD and KELIM

The discussion is insufficient and does not discuss the value of this model in depth. 

Minor points

Abstract: KPROD and KELIM are not standard acronyms. Needs clarification. I am not familiar with these concepts and thus they probably need to be introduced in the abstract prior to use

Introduction: need to cite GETUG-15, STAMPEDED and CHAARTED

Figure 1: what is K-PD?

Reviewer 2 Report

The present study is submitted for the Special Issue "Biomarkers for Detection and Prognosis of Prostate Cancer", aiming for the investigation of biomarkers for detection and prognosis of prostate cancer to especially improve risk stratification between low- and high risk of biochemical recurrence and disease progression.

The study adequately addresses this topic by assessing longitudinal kinetics of PSA in association with patient prognosis during androgen-deprivation therapy +/- docetaxel of patients with rising PSA-level after primary local therapy. The significance of improved risk stratification of low- and high risk of biochemical recurrence is raised and a mathematical modeling approach to overcome existing obstacles in biomarker evaluation is presented. The thematic classification shows clinical significance and the implementation of a mathematical model in regularly assessed PSA values demonstrates an interesting alternative to standard evaluation.

Abstract and Single Summary adequately outline the main aspects and results of the study. However, the clinical background and patient selection could be pointed out more clearly.

The Introduction illustrates the clinical background and thematic classification, including its significance and the need for improved risk stratification. Therapeutical options for locally treated patients with risk for metastatic diseases and for already metastasized tumors are presented. However, the current State-of-the-art could be pointed out with more references, and a few more recent papers could be cited (reference 1 and 2 were published in 2003 and 2004).

Within the section Material and Methods, the study design as well as criteria for inclusion and exclusion are reasonably listed. It is correctly revealed that patients harboring stable or rising PSA values within the given timeframe were excluded from the analysis. This aspect should again be mentioned in the discussion to outline that the resulting prognostic value of the PSA kinetics is related to a basically decreasing PSA-level, whereas (initially) stable and increasing PSA-levels were not considered in the association to PSA-PFS and OS. It is equally explained, why a few patients were dropped after inclusion and therefore are not included in the final analysis. This procedure is comprehensive, but could nevertheless result in unobvious statistical bias.

The Results are appropriately presented and include patient selection, model qualification and comprehensive description of results in statistical calculations, supported by two Figures and one table. Figure 2 clearly represents the framework of the study. Within the part Patient selection, it was referred to Supplementary Table S1, summarizing the characteristics of selected and excluded patients. As stated, the characteristics are similar in both treatment arms, concerning age, stage T3/T4 and Gleason score > 8. Interestingly, in case of excluded patients, the ratio of RP as prior treatment compared to RT as prior treatment is higher than in case of selected patients. Equally, PSA levels are halved in excluded patients compared to selected patients, whereas all other characteristics are indeed similarly distributed. It would be interesting to comment these aspects.

Finally, the Discussion summarizes statistical results and refers to clinical relations. Limitations of the study are briefly discussed. However, the discussion is rather short and clinical significance as well as limitations and comparison with previous work should be further expanded. Relating to limitations of the study, although being basically advantageous, serial assessments of the PSA level could underly fluctuation and biasing aspects, as well. A prospective validation of the results would be helpful. Equally, it was not mentioned whether serial PSA assessments were performed in comparable time periods between different patients and between the two treatment arms, which could further influence the results.

Within the Introduction it was reasonably illustrated that subgroups could benefit from an appropriate risk stratification. This aspect was only mentioned in the Discussion shortly by stating that no significant differences were seen between the two treatment arms. The clinical relation and significance should be further taken up in the Discussion: how could the presented approach and the prognostic value of KELIM and KPROD reach clinical benefit?

Formal aspects:

  • Table 1 (main text): Layout in table A should be adapted to layout in table B.
  • Supplementary Table S1: It appears as if the total number of patients for ADT+Docetaxel and ADT alone in case of excluded patients was wrongly assigned, when comparing the numbers to the part Results, Patient selection in the main text. Consequently, the presented percentages in Age, Stage T3 /T4, Prior RP and Prior RT are also wrong. The total numbers or patients with prior RP or prior in case of both, selected patients as well as excluded patients (ADT+Docetaxel and ADT alone), are irritating since the sum of prior RP plus prior RT does not equal to the total patient number in the respective group.

Round 2

Reviewer 1 Report

I thank the authors for addressing my concerns in the previous iteration of the manuscript. I think this work has been strengthened considerably and is now suitable for publication

Reviewer 2 Report

The manuscript was substantially improved. We congratulate the authors for their work.